# An Efficient Deep Learning Approach to Automatic Glaucoma Detection Using Optic Disc and Optic Cup Localization

**DOI:** 10.3390/s22020434

**Published:** 2022-01-07

**Authors:** Marriam Nawaz, Tahira Nazir, Ali Javed, Usman Tariq, Hwan-Seung Yong, Muhammad Attique Khan, Jaehyuk Cha

**Affiliations:** 1Department of Computer Science, University of Engineering and Technology Taxila, Rawalpindi 47050, Pakistan; marriam.nawaz@uettaxila.edu.pk (M.N.); tahira.nazir77@gmail.com (T.N.); ali.javed@uettaxila.edu.pk (A.J.); 2Information Systems Department, College of Computer Engineering and Sciences, Prince Sattam Bin Abdulaziz University, Al Khraj 11942, Saudi Arabia; u.tariq@psau.edu.sa; 3Department of Computer Science and Engineering, Ewha Womans University, Seoul 03760, Korea; hsyong@ewha.ac.kr; 4Department of Computer Science, HITEC University, Rawalpindi 47080, Pakistan; 5Department of Computer Science, Hanyang University, Seoul 04763, Korea; chajh@hanyang.ac.kr

**Keywords:** fundus images, glaucoma, EfficientDet, EfficientNet

## Abstract

Glaucoma is an eye disease initiated due to excessive intraocular pressure inside it and caused complete sightlessness at its progressed stage. Whereas timely glaucoma screening-based treatment can save the patient from complete vision loss. Accurate screening procedures are dependent on the availability of human experts who performs the manual analysis of retinal samples to identify the glaucomatous-affected regions. However, due to complex glaucoma screening procedures and shortage of human resources, we often face delays which can increase the vision loss ratio around the globe. To cope with the challenges of manual systems, there is an urgent demand for designing an effective automated framework that can accurately identify the Optic Disc (OD) and Optic Cup (OC) lesions at the earliest stage. Efficient and effective identification and classification of glaucomatous regions is a complicated job due to the wide variations in the mass, shade, orientation, and shapes of lesions. Furthermore, the extensive similarity between the lesion and eye color further complicates the classification process. To overcome the aforementioned challenges, we have presented a Deep Learning (DL)-based approach namely EfficientDet-D0 with EfficientNet-B0 as the backbone. The presented framework comprises three steps for glaucoma localization and classification. Initially, the deep features from the suspected samples are computed with the EfficientNet-B0 feature extractor. Then, the Bi-directional Feature Pyramid Network (BiFPN) module of EfficientDet-D0 takes the computed features from the EfficientNet-B0 and performs the top-down and bottom-up keypoints fusion several times. In the last step, the resultant localized area containing glaucoma lesion with associated class is predicted. We have confirmed the robustness of our work by evaluating it on a challenging dataset namely an online retinal fundus image database for glaucoma analysis (ORIGA). Furthermore, we have performed cross-dataset validation on the High-Resolution Fundus (HRF), and Retinal Image database for Optic Nerve Evaluation (RIM ONE DL) datasets to show the generalization ability of our work. Both the numeric and visual evaluations confirm that EfficientDet-D0 outperforms the newest frameworks and is more proficient in glaucoma classification.

## 1. Introduction

Glaucoma is a malicious eye disease that harms the eye’s optic nerve because of the usual intraocular pressure (IOP) in it [1]. The difference in the produced and drained range of intraocular fluid (IOF) of the eye results in IOP which in turn affects the nerve fibers (NF). The damaged NF disturbs the retinal nerve fiber layer (RNFL) and causes to increase in the cup-to-disc ratio (CDR) (or “cupping”) and optic disc (OD) or optic nerve head (ONH) [2]. Furthermore, the IOP also causes to weaken the retinal pigment epithelium namely peripapillary atrophy (PPA). Existing research work has confirmed that a rise in the growth of PPA results in generating acceleration in glaucoma [3]. A sample of glaucomatous eyes is presented in Figure 1, from where it can be visualized that the blockage in IOF damages the optic nerve. Furthermore, it can be seen that the OD volume for the glaucoma-affected eye is larger in comparison to a normal human eye.

In a recent report, it is stated that glaucoma is the main cause of blindness in people and its growth rate is increasing exponentially which tends to affect 80 million humans by 2021 all around the world [4]. The advanced stage of glaucoma can cause the complete vision loss of the victim and it is usually identified at its severe level. Because of such reasons, glaucoma is given the name of “silent thief of sight” [5]. Even though extensive advancements have been introduced in the area of medical image analysis [6,7,8,9]. However, the accurate localization and treatment of glaucoma-affected areas are incurable. Whereas timely detection of this drastic disease can save the victims from complete sightlessness. In another study conducted in [10], it is forecasted that by 2040, the number of glaucomatous victims will rise to 111.8 million. The extensive increase in the growth rate of glaucoma will introduce a social and financial load on the world economy and have an impact on the comfort of victims [1].

At the start, manual eye grading systems were used by the ophthalmologists through visually examining the CDR and OD area to locate the irregularities of border areas. However, the increase in the number of victims and dependency on the availability of experts often delayed the diagnostic process which in turn increased the cases of complete vision loss [10]. To tackle the needs of a large population, the research community initiated the step of introducing fully automated glaucoma recognition approaches. Usually, the IOP measurement is used to identify the various eye-related diseases where the previous health history of victims is used, and eye field loss tests are conducted by ophthalmologists to visually examine the structure, size, and color of the optic nerve. For that reason, accurate localization and segmentation of the glaucomatous area is not only necessary for better eyes medical examinations by ophthalmologists but also required for designing a fully automated system for effective disease classification which is prone to less error rate [9]. Initially, the hardcoded feature-based approaches have been employed by the researchers to discriminate the healthy and affected regions of human eyes [11,12]. However, these approaches work by first locating the region of interest (ROI), which in turn increases the economic cost of CAD systems and is not much effective in glaucoma recognition due to extensive changes in the attributes of lesions [13]. Now, the effectiveness of DL-based methods has grabbed the attention of researchers to employ them in the area of medical image analysis [14,15,16]. The DL-based approaches are capable of automatically extracting the representative set of image features without requiring the assistance of experts and obtaining better performance with small preprocessing and computation power. Moreover, DL-based methods are robust to localize the lesions of varying sizes by examining the topological features of suspected samples and are more reliable to deal with the various image distortions like size, rotation, and scale variations of glaucoma-affected regions.

Even though a huge amount of work has been presented for the automated localization and classification of glaucoma lesions, however, there is a need for performance enhancement. Although, the power of ML approaches to better tackle tough real-life scenarios is significant in comparison to the humans’ intelligence. However, these techniques may not perform well for the samples post-processing attacks and show high computation complexity as these methods generate long codes that rise the processing time. To tackle the problems of ML approaches, the DL-based frameworks are utilized, however, they increase the code complexity. Furthermore, these methods are not well-suited to everyday problems because of the changing attributes of the glaucoma-affected areas. Therefore, there is a need to both improve the detection accuracy and processing time for glaucoma-affected regions identification and classification.

The challenging nature of glaucoma lesions like the intense variations in their size, color, and structure has made them challenging to be diagnosed at the earliest stage. To deal with the aforementioned challenges, a DL-based technique named EfficientDet [17,18] is introduced with Efficient-B0 as a backbone architecture. In the first step, the Efficient-B0 feature computation unit of EfficientDet-D0 is utilized to calculate the deep key points. Then, the computed keypoints are identified and categorized through the one-stage detector of EfficientDet-D0. For performance analysis, we have used two standard datasets namely ORIGA and HRF, and validated through the obtained results that the presented framework provides an effective and efficient solution to glaucoma lesion classification under the occurrence of extreme alterations in volume, color, and texture of lesions. Moreover, the EfficientDet-D0 is also robust to glaucomatous region recognition under the occurrence of intensity changes, noise, and blurring in the suspected samples. Following are the main contributions of our work:We present a robust model namely EfficientDet-D0 with EfficientNet-B0 for keypoints extraction to enhance the glaucoma recognition performance while decreasing the model training and execution time.The presented technique can accurately identify the glaucomatous regions from the human eyes because of the robustness of the EfficientDet framework.Accurate detection and classification of glaucoma-affected images due to the ability of the EfficientDet model to tackle the over-trained model data.The model is computationally robust as EfficientDet uses a one-stage object identification procedure.Huge performance evaluations have been performed over the two datasets namely ORIGA and HRF which are diverse in terms of varying lesion color, size, and positions and contain samples with several distortions to show the robustness of the proposed solution.

The remaining manuscript follows the given distribution: Section 2 contains the related work, whereas the presented approach is explained in-depth in Section 3. Section 4 contains the obtained results together with the details of the employed database and evaluation metrics while the conclusion is drawn in Section 5.

## 2. Related Work

In this work, we have discussed the work from the history employed for the detection and classification of glaucoma lesions from the fundus samples. The methods used for glaucoma recognition are classified either as ML-based approaches or DL-based techniques.

Shoba et al. [19] introduced an ML-based method for glaucomatous region detection. After performing the preprocessing step, the Canny Edge Detection (CED) approach was applied to perform the blood vessels segmentation. Then the morphological operation was performed for segmenting the blood vessels from the suspected sample. In the next step, the Finite Element Modeling (FEM) analysis was conducted for final feature computation. The computed features were used for the support vector machine (SVM) training to perform the classification task. The work [19] is robust to noisy samples, however, the model needs to be evaluated on a challenging dataset. In [20] a method namely the Glowworm Swarm Optimization algorithm was introduced for the automated identification of optic cups from retinal fundus samples. The framework [20] is robust to glaucoma detection, however, unable to compute the cup-to-disc ratio. Kirar et al. [21] presented an approach for glaucoma identification employing second-stage quasi-bivariate variational mode decomposition (SS-QB-VMD)-based fine sub-band images (SBIs) from suspected samples. The computed features from the SS-QB-VMD framework were used to train the least-squares SVM (LS-SVM) classifier. The work [21] performs well for glaucoma detection, however, classification accuracy needs further improvements. Qureshi et al. [22] presented a framework to recognize the glaucomatous lesions. After performing the image preprocessing task, the OD and OC were segmented by employing the using pixel-based threshold and watershed transformation approaches. Finally, the CDR was computed by distributing the number of cup pixels by the number of disc pixels. The work [22] performs well for the glaucomatous region recognition, however, may not perform well for the scale and rotation variations in the suspected samples. In [23] an ML-based automated framework was presented to calculate the vertical cup-to-disk ratio (VCDR) to identify the glaucomatous areas from the fundus images. Initially, the vasculature and disk selective COSFIRE filters were employed for OD localization. After this, a generalized matrix learning vector quantization (GMLVQ) classifier was utilized for classifying the OD and OC regions. The work shows better glaucoma detection accuracy, however, not robust to noisy samples.

Martins et al. [24] presented an approach by introducing a lightweight CNN framework for glaucoma recognition. After performing the preprocessing step, the MobileNetV2 approach was used to compute the deep features from the input images which were later classified as healthy, and glaucoma affected. The work is computationally better, however requires extensive data for model training. In [25] another DL-based approach was introduced for the automated classification of glaucoma-affected samples from the healthy images. A framework namely evolutionary convolutional network (ECNet) was introduced for reliable keypoints extraction from the input images. After this, the extracted key points were employed for training the several ML-based classifiers i.e., K-nearest neighbor (KNN), SVM, backpropagation neural network (BPNN), and extreme learning machine (ELM) to perform the classification task. The work obtains the best results with the SVM classifier, however, at the charge of the enhanced processing burden. Shinde et al. [26] introduced a DL-based framework for the automatic detection and categorization of glaucoma from the input samples. Initially, the Le-Net architecture was used to identify the Region of Interest (RoI) from the input images. Then the U-Net framework was used to execute the OD and OC segmentation. Finally, the classification task was performed by employing the SVM, NN, and Adaboost classifiers. The work [26] attains better accuracy by combing the SVM, NN, and Adaboost classifiers results, which in turn increase the computational cost. Song et al. [27] presented a CNN-based framework in which the Design of experiments (DOE) analysis was performed for attaining robust hyperparameters. The work [27] shows better glaucoma classification performance, however, the framework needs evaluation on some standard datasets. In [28], another approach namely ResNet-50 was used to identify and recognize the glaucomatous regions from the fundus images. The work presented in [28] demonstrates improved glaucoma detection results, however, may not be robust to the noisy and blurred images. Similarly, in [29] another DL-based framework namely DenseNet-201 was presented for the automated recognition of glaucoma. The approach [29] is computationally better, however, performance needs further improvements. Serte et al. [30] introduced an ensemble technique for OD and OC recognition. The deep features from three models namely AlexNet, ResNet-50, and ResNet-152 were fused to predict the healthy and glaucoma affected regions. The work [30] shows better glaucoma classification performance, however, this framework is computationally expensive. Nazir et al. [31] introduced a methodology namely Mask-RCNN to cluster OD and OC lesions from the fundus samples. Initially, DenseNet-77 was applied as a backbone in the Mask-RCNN to extract the deep key points from the input image which were later segmented by the Mask-RCNN framework. The method [31] performed well to glaucoma segmentation, however, segmentation results need more improvements. Similarly, in [32] another DL-based approach namely Fast Region-based Convolutional Neural Network (FRCNN) algorithm with fuzzy k-means (FKM) clustering was introduced. The approach [32] exhibits better glaucoma segmentation performance, however, at the expense of large economic costs. Yu et al. [33] introduced a DL-based technique to detect glaucoma by changing the U-net framework by replacing the down-sampling encoding layers with the ResNet-34 framework. This work [33] exhibits better glaucomatous recognition accuracy, however, detection accuracy is dependent on the quality of fundus samples. In [34] a VGG19 framework by using the concept of transfer learning was applied to detect glaucoma from the suspected images. This technique works well for glaucoma detection, however, needs extensive data for model training. Bajwa et al. [35] introduced a two-stage network to identify and classify the glaucomatous areas from the input images. Initially, the Faster-RCNN model was used to localize the ROI (optic disc) which were later classified by the CNN classifier. This work performs well in comparison to the heuristic localization approaches, however, not robust to extensive color variations in the input images. Moreover, in [36] a weakly supervised multi-task learning (WSMTL) approach was presented for the automated identification and classification of glaucoma. The CNN feature extractor containing skip connections was used to calculate the deep key points from the input images which were later classified to healthy and glaucoma-affected images. The approach [36] is computationally robust, however, classification performance needs further improvements. Another similar approach was introduced in [37] where the ResNet framework with multi-layers average pooling was used to perform the mapping among the global semantic information and precise localization. The approach shows better glaucoma detection accuracy; however, the model may not perform well for blur images.

An analysis of existing techniques used for glaucoma recognition is provided in Table 1. From Table 1, it can be seen that still, there exists a demand for a more robust framework that can present both effective and efficient results for glaucomatous region classification.

## 3. Proposed Methodology

The presented approach comprises two steps: (i) data preparation (ii) glaucoma detection and categorization step. The main flow of the presented solution is exhibited in Figure 2. In the data preparation step, we develop annotations by drawing a bounding box (Bbox) to exactly locate the RoIs. Secondly, the EfficientDet framework is trained over the annotated images to recognize glaucoma-affected regions. We used EfficientDet-D0 with EfficientNet-B0 as its base network for features extraction. The EfficientDet-D0 follows three steps to localize and classify glaucoma-affected regions. In the first step, the keypoints calculator of the EfficientDet-D0 network namely EfficientNet-B0 takes two types of input (suspected image and annotations). In the next step, the BiFPN module performs the top-down and bottom-up keypoints fusion several times for the resultant features of Level 3–7 in EfficientNet. In the third step, the final localized region with the associated class is predicted and results are computed for all modules as per evaluation parameters being used in the area of computer vision. Algorithm 1 specifies the in-depth explanation of the introduced technique.
**Algorithm 1: Steps for the presented method.****INPUT:**TrD, Ann**OUTPUT:**Localized RoI, EfficientDet, Classified glaucoma diseased portion***TrD*—**training data.***Ann*—**Position of the glaucomatous region in suspected images.**Localized RoI—**Glaucomatous area in output.***EfficientDet*—**EfficientNet-B0 based EfficientDet network.**Classified glaucoma diseases portion—**Class of identified suspected region.**imageSize ← [x y]****Bbox calculation****      *µ* ← AnchorsCalculation (*TrD*, Ann)****EfficientDet—Model**     ***EfficientDet** ←* EfficientNet-B0-Based EfficientDet (*imageSize*, *µ*)      **[*d_r_ d_t_*]** ← Splitting database in the *training* and *testing set***The training module of glaucoma recognition****For** each sample ***s*** in → ***d_r_*****Extract***EfficientNet-B0-*keypoints → *ds***Perform features Fusion** (*ds*) → *Fs***End**Training ***EfficientDet*** on *Fs*, and compute processing time *t_Edet****η_Edet* ←** DetermineDiseasedPortion(*Fs*)***Ap_ Edet* ←** Evaluate_AP (*EfficientNet-B0*, *η_ Edet)***For** each image ***S*** in → **d_t_**(**a**) Calculate key points via trained network € → β*I* (**b**) [*Bbox*, *localization_score*, *class*] ← Predict (β*I*) (**c**) Output sample together with *Bbox*, *class*(**d**) *η* ← [*η Bbox*]**End For*****Ap_*€** ← Evaluate model € employing *η****Output_class*** ← EfficientDet (*Ap_*€).

### 3.1. Annotations

For an accurate and correct training procedure, it is essential to precisely demonstrate the position of the glaucoma-affected areas from the suspected samples. To accomplish this task, we have employed the LabelImg [26] software to generate the annotations of affected image areas to exactly specify the RoIs. Figure 3 presents some of the generated annotations. The developed annotations are saved in an XML file which carries two types of information: (i) coordinate values of generated Bbox on the glaucomatous area (ii) class associated with each detected region. Then, the training file is generated from the XML file which is further employed for network training.

### 3.2. EfficientDet

Efficient and effective feature extraction is necessary to correctly classify the suspected samples as glaucoma-affected or healthy images. At the same time, obtaining a more representative set of image features is a complex job because of the following causes: (i) the computation of a larger-sized feature vector can cause the framework to result in a model over-fitting problem and (ii) whereas, a small-sized feature vector can cause the framework to miss to learn some essential sample aspects like color and texture changes which make diseased parts of an image indistinguishable from the healthy areas. To have a more representative set of image keypoints, it is essential to use an automatic keypoints calculation approach without employing hand-coded features computation method. The frameworks utilize hand-coded features which are not effective in precisely locating and classifying glaucomatous regions due to huge variations in the size, structure, chrominance, position, and subtle border of glaucoma lesions. To tackle the aforementioned issues, we utilized a DL-based approach namely EfficientDet [17,18] due to its power to automatically extract the robust key points from the samples under investigation. The convolution filters of EfficientDet calculate the features of the input sample by investigating its structure. Several object detection methods have been presented by the researchers for the localization and recognition of medical diseases. These detectors are classified either as one-stage (YOLO, SSD, RetinaNet, CornerNet, CeneterNet) or two-stage (RCNN [38], Fast-RCNN [39], Faster-RCNN [40], Mask-RCNN) object detectors. The motivation of selecting EffieicntDet in comparison to other one-stage detectors is that these methods compromise the classification accuracy by showing a minimum time to perform the classification task. While the two-stage detectors exhibit better lesion detection accuracy, however, at the charge of enlarged processing complexity as these techniques perform two steps to locate and classify the ROIs and which makes them unsuitable for real-world scenarios. Therefore, there is a need to represent such an approach that will give a vigorous and effective solution to glaucoma lesion recognition and categorization.

To overcome the above-mentioned issues, we have used the EfficientDet approach which was presented by the Google brain team. By enhancing the multi-directed keypoints fusion architecture of FPN and by deriving the idea from the EfficientNet framework scaling approach for reference, the EfficientDet model is a scalable and robust object identification algorithm. The EffificientDet approach comprises three main modules, the first part is EfficientNet which is the feature extractor module. In our work, we have used EfficientNet-B0 as the base network to calculate the reliable keypoints from the input images. The second module is named BiFPN, which performs both top-down and bottom-up keypoints fusion several times for the resultant feature vector of Level 3–7 in EfficientNet. And the last module is used to localize and classify the detected region as glaucomatous affected or healthy. The detailed description of training parameters used by the EfficientDet is given in Table 2.

The detailed description of all three modules is given as:

#### 3.2.1. Feature Extraction through EfficientNet-B0

We have used EfficientNet-B0 as a base network for extracting the deep features from the suspected samples. In comparison to traditional methods that randomly scale network dimensions, i.e., width, depth, and resolution, the EfficientNet method consistently scales each dimension with a fixed set of scaling coefficients. The EfficientNet-B0 is capable of computing the more representative set of image features with a small number of parameters which in turn improves the detection accuracy by minimizing the computation time as well. Figure 4 presents the structure of the EfficientNet-B0 framework. The EfficientNet framework is capable of presenting the complex transformation accurately which enables it to better deal with the issue of the absence of the ROIs position information. Additionally, the EfficientNet framework allows reusing the computed features which make it more suitable for glaucoma disease identification and fasten the training procedure

#### 3.2.2. BiFPN

In glaucoma detection and classification application, key points like lesion position, background, light variations, and the affected region size must be taken into consideration. Therefore, utilizing multi-scale keypoints computation can assist in accurately recognizing the glaucomatous regions. In history, frameworks usually employ the top-down FPNs to fuse the multiscale keypoints However, in the one-sided FPN, varying scales are not essentially participated equally to the resultant features which can result in missing to learn some important image behaviors in glaucoma detection procedures. Therefore, in the presented approach the concept of BiFPN is introduced to better tackle the problem of equal contribution in FPN. The BiFPN module allows information to flow in both the top-down and bottom-up directions via employing regular and reliable connections. Moreover, the BiFPN module uses trainable weights to extract semantic-based keypoints having significant contributions to the resultant framework. Therefore, key points from P3 to P7 layers of the EfficientNet-B0 are nominated as multi-scale features and passed as input to the BiFPN module. The width of the BiFPN module upgrades exponentially as the depth increases linearly, and must have to satisfy the given Equation (1):(1)$$W_{bf}=64.\;(1.35^{\emptyset }),\;D_{bf}=3+\emptyset$$

Here, $W_{bf}$ and $D_{bf}$ are presenting the width and depth of the BiFPN module, respectively, while ∅ is the compound factor that controls the scaling dimensions which is 0 in our case.

#### 3.2.3. Box/Class Prediction Network

The combined multi-scaled key points from the BiFPN module are passed to Box/class prediction module to draw a Bbox across the suspected region and specify the associated class. The width of this module is the same as that of the BiFPN, however, depth is computed by using Equation (2):(2)$$D_{Bbox}=3+\left[\emptyset /3\right]$$

### 3.3. Detection Procedure

The EfficientDet approach is free from approaches like selective search and proposal generation. Therefore, the input samples along with the generated annotations are feed to the EfficientDet approach, on which it directly computes the lesion position along with the dimensions of Bbox and associated lesion class.

## 4. Experimental Results

In this section, we have discussed the detailed analysis of acquired results after conducting several experiments to compute the glaucoma identification and categorization power of the introduced framework. Moreover, we have discussed the details of employed databases and evaluation metrics as well.

### 4.1. Dataset

To check the robustness of our approach for glaucoma detection and classification, we have used a publically accessible database namely ORIGA [41]. The ORIGA database comprises 650 samples, where 168 images contain the glaucoma-affected regions, while the remaining 650 images are from the normal human eyes. The ORIGA dataset is a challenging dataset for glaucoma classification as its samples contain several artifacts for example huge variation in the size, color, position, and texture of OD and OC. Moreover, images contain several distortions like the presence of noise, blurring, color, and intensity variations. Samples from the employed dataset are shown in Figure 5.

### 4.2. Evaluation Metrics

In this work, several assessment measures i.e., Intersection over Union (*IoU*), accuracy, precision, recall, and mean average precision (*mAP*) are used to check the localization and categorization performance of our approach [42]. Accuracy is measured by using Equation (3).
(3)$$Accuracy=\frac{TP+TN}{TP+FP+TN+FN}$$

Equation (4) demonstrates the calculation of the *mAP* score, where *AP* is showing the average precision from all classes, while q is denoting the sample under the test. Moreover, Q is denoting the total test samples.
(4)$$mAP:=\sum_{i=1}^{T}AP(t_{i})/T$$

Equations (5)–(7) show the *IoU*, precision, and recall, respectively.
(5)$$IoU=\frac{TP}{FN+FP+TP}\times 2$$
(6)$$\mathrm{Precision}=\frac{TP}{TP+FP}$$
(7)$$\mathrm{Recall}=\frac{TP}{TP+FN}$$

### 4.3. Proposed Technique Evaluation

Timely and precise identification of the OD and OC lesions is mandatory for designing an effective computer-aided approach for glaucoma-affected regions identification and classification. For this reason, we have designed an experiment to assess the localization ability of EfficientDet by checking its recognition power on all test samples from the ORIGA database, and obtained outputs are shown in Figure 6. It is clearly visible from the reported results that the proposed solution namely EfficientDet is capable of diagnosing the OD and OC lesions of varying sizes and positions. Moreover, our work is capable of dealing with numerous samples distortions like blurring, color, and brightness variations.

The localization ability of the EfficientDet approach permits it to precisely recognize the lesions exhibiting fewer signs. Furthermore, for the quantitative estimation of our approach, we have utilized two evaluation measures namely *mAP* and *IoU*, as these measures are the most widely employed by the researchers and assist in better evaluating the localization power of a system. Our approach obtains an average *mAP* and means *IoU* values of 0.971 and 0.981, respectively. It can be seen from both the visual and numerical results that our framework is reliable to localize and categorize the glaucoma-affected regions.

Moreover, for robust glaucoma detection and classification framework, it must be capable of differentiating the glaucomatous samples from the healthy images. For this reason, we have plotted the confusion matrix as it can better demonstrate the classification results by showing the true positive rate (TPR). The obtained results are shown in Figure 7, from where it can be witnessed that for glaucoma-affected images, the EfficientDet approach shows a TPR of 0.970 which is clearly showing the effectiveness of our approach. Furthermore, our technique acquires an average glaucoma classification accuracy of 97.2% on the ORIGA dataset. The main reason for the robust classification accuracy of our method is that EfficientDet with EfficientNet-B0 as the base network is capable of computing the more accurate set of image features which better assist in diagnosing the diseased image areas.

### 4.4. Comparison with Other Object Detection Approaches

We have experimented to evaluate the glaucoma recognition results of our framework with other object detection approaches i.e., RCNN, Faster-RCNN, and Mask-RCNN. The obtained analysis is shown in Table 3. To perform the comparative analysis with other object detection techniques, we have considered the *mAP* evaluation metric, as it is designated as a standard by the research community in object recognition systems. Furthermore, we have compared the models testing time to assess these approaches in terms of computational burden as well. From Table 3, it is clear that our framework attains the highest *mAP* value of 0.971, along with the smallest testing time of 0.20. Furthermore, the RCNN approach attains the lowest *mAP* value of 0.913 and has the largest testing time of 0.30 as well. Moreover, the Mask-RCNN approach with DenseNet-77 shows comparable results to our work, however, it is computationally more expensive because of its two-step feature locator architecture. Hence, it is noticeable that our work is more effective in glaucoma lesion detection and classification due to its one-stage object detection ability which provides it the computational benefit on the other techniques. Moreover, the reliable feature detection by EfficientNet-B0 enables the EfficientDet-D0 framework to accurately localize the ROIs and attain the highest *mAP* value to its competitors

### 4.5. Comparison with State-of-the-Art

To further check the glaucoma identification and classification performance of our approach, we have conducted another analysis in which the latest approaches employing the same dataset are chosen for comparison. To have a fair analysis, we have taken the average results of our technique and have evaluated them with the average results of approaches in [31,32,35,37,43]. The comparative quantitative results with the help of standard evaluation metrics are shown in Table 4.

Liao et al. [37] presented a DL-based framework for glaucoma recognition from retinal fundus samples and gained an average AUC of 0.88. Fu et al. [44] also proposed a DL-based framework namely Disc-aware Ensemble Network (DENet) to identify and classify the glaucomatous samples with the average AUC and Recall values of 0.901 and 0.920 respectively. Moreover, the work in [35] presented a Two-stage framework for OD and OC detection to classify the glaucoma-affected images and showed the average AUC and Recall values of 0.0.868 and 0.710, respectively. Nazir et al. [32] proposed an approach namely Fast Region-based Convolutional Neural Network (FRCNN) and acquired the average AUC and Recall of 0.941 and 0.945 respectively. Similarly, Nazir et al. [31] proposed a DL framework namely Mask-RCNN to recognize the glaucomatous regions from the retinal samples and attained an average AUC and Recall of 0.970 and 0.963 respectively. Whereas the presented framework namely EfficientDet-D0 with EfficientNet-B0 as base network obtain the average AUC and Recall values of 0.979 and 0.970, which are higher than all the comparative approaches. More specifically, for the AUC evaluation metric, the competitive approaches acquire an average value of 0.0.9138, whereas the presented approach shows the AUC value of 0.979, so, EfficientDet-D0 framework gives a performance gain of 6.52%. While in the case of Recall, the comparative approaches show an average value of 0.8845, which is 0.970 for our approach. Therefore, we obtain the performance gain of 8.55% for Recall and clearly demonstrate the robustness of EfficientDet-D0 for glaucoma classification. Moreover, we have compared the proposed solution with other approaches in terms of time complexity as well. It can be seen from Table 4 that our work shows minimum time in comparison to all other methods due to its one-stage detection power.

The main reason for the better performance of our approach in comparison to other techniques is that these methods [31,32,35,37,43] employ very complex and deep networks for feature computation, which eventually cause the model over-fitting problem and increase the computational complexity of models. Whereas, in comparison, our approach employs EfficientNet-Bo as a base network, which is capable of computing the more representative set of image features while maintaining the computational complexity as well. Hence, it can be concluded that EfficientNet-Bo-based EfficientDet-D0 architecture provides an efficient and effective solution to OD and OC recognition which can assist the doctors to timely diagnose the glaucoma-affected regions.

### 4.6. Cross Dataset Validation

We conducted an analysis via evaluating the proposed solution over a cross-dataset on namely HRF [42]. This database contains 45 samples, of which 15 images are from the healthy human eye, while 15 images contain DR-affected regions, and the remaining 15 samples are glaucomatous-affected. The performance analysis on a cross-dataset assists to check the recognition accuracy of our approach in terms of its generalization ability to real-world examples. Moreover, this experiment will help to determine whether our system is capable of dealing with the training and testing complexities. More explicitly, we have trained our system on the ORIGA dataset and tested it on the HRF database. The obtained results are shown in Figure 8 by plotting the Boxplot, as it better demonstrates the performance of the system by exhibiting the maximum, minimum, and median of the acquired accuracies. In this experiment, our approach shows the average accuracies of 98.98% and 98.21% for the training and testing respectively which is concluding that our framework can be employed in real-world problems to cope with the challenges of OD and OC recognition and can better assist the ophthalmologist in the early diagnosis of glaucoma.

We have further evaluated our method on a challenging dataset named: RIM ONE DL [44,45] which is the latest version of the RIM ONE dataset. This dataset consists of 485 images of which 313 from normal and 172 are images of the patients affected from glaucoma. We have performed two types of experiments to further check the generalization ability of our approach and results are reported in Figure 9 and Figure 10. In the first experiment, we have trained the model on the ORIGA dataset and test it on the RIM ONE DL, and obtained the average train and test accuracy of 98.91% and 97.96%, respectively. For the second evaluation, we have trained the proposed framework on the RIM ONE DL dataset while evaluating it on the ORIGA database and attain the average accuracy values of 98.14% and 97.83%, respectively. It can be seen from the reported results in Figure 9 and Figure 10 that our work is capable of robustly classifying the unseen examples.

We have conducted the cross-dataset evaluation on different challenging datasets namely the ORIGA, HRF, and RIM-ONE DL datasets. The ORIGA dataset is more challenging and large-sized in compassion to the other two databases. While the RIM-ONE DL dataset is complex in nature than the HRF dataset. We have performed a comparative analysis of cross-dataset validation and obtained results are reported in Table 5. It is quite clear from Table 5 that our work has acquired reliable performance on all databases and is robust to classify the unseen images efficiently.

## 5. Conclusions

The manual recognition of the glaucomatous-affected regions from fundus samples requires trained human experts who can identify the small visible details and categorize the images into relevant classes. However, because of the complex structure of the glaucomatous regions and the unreachability of domain experts, there is a need for a fully automated system. In the introduced technique, we have presented a DL-based approach named EfficientDet-D0 with EfficientNet-B0 as the base network for the automated localization and categorization of glaucoma lesions from the retinal fundus images. We have tested our approach over the ORIGA database which is challenging in terms of variations of glaucoma lesion size, color, position, and shapes. Moreover, to assess the generalization ability of our framework to real-world problems, we perform cross-dataset validation on the HRF and RIM ONE DL datasets. For the ORIGA database, we obtain the average accuracy values of 97.2%, while for the HRF and RIM ONE DL databases, we obtain an average accuracy of 98.21% and 97.96% respectively. Both the visual and numeric comparison confirms that the used framework is more robust to glaucoma classification as compared to other latest approaches and can certainly identify the lesions of variable masses from the samples with several image distortions. Therefore, this work can play a vital role in the automated recognition and classification of the glaucomatous-affected regions. In the future, we plan to implement some feature section techniques and employed on deep learning models [3,46,47,48,49,50]. Also our plan is to evaluate the work on other eye diseases.

## Figures and Tables

**Figure 1 sensors-22-00434-f001:**
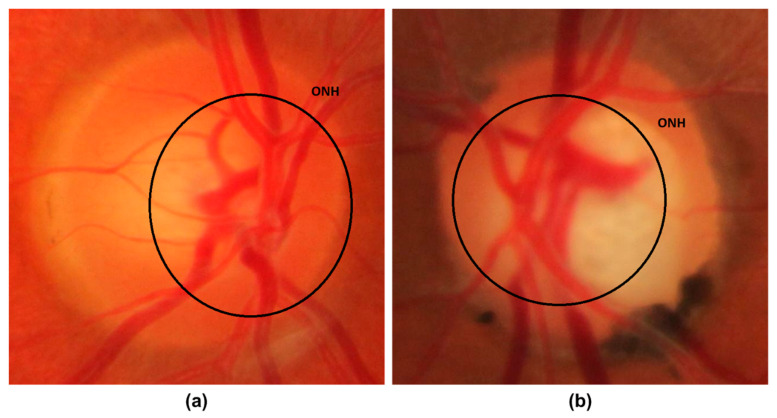
Optic Nerve Head images (**a**) Normal eye (**b**) Glaucomatous eye image.

**Figure 2 sensors-22-00434-f002:**
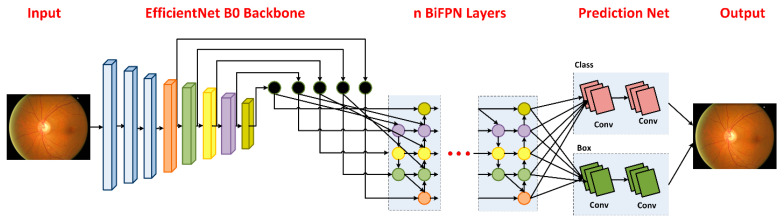
Flow diagram of Proposed Technique.

**Figure 3 sensors-22-00434-f003:**
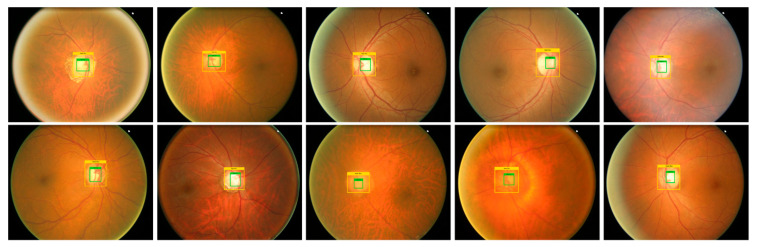
Annotation samples.

**Figure 4 sensors-22-00434-f004:**
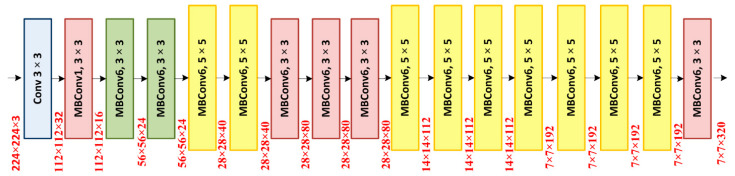
EfficientNet-B0 architecture.

**Figure 5 sensors-22-00434-f005:**
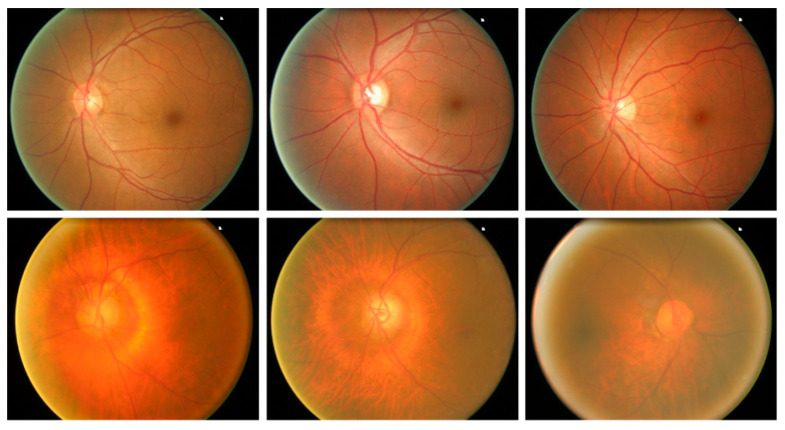
Sample dataset images.

**Figure 6 sensors-22-00434-f006:**
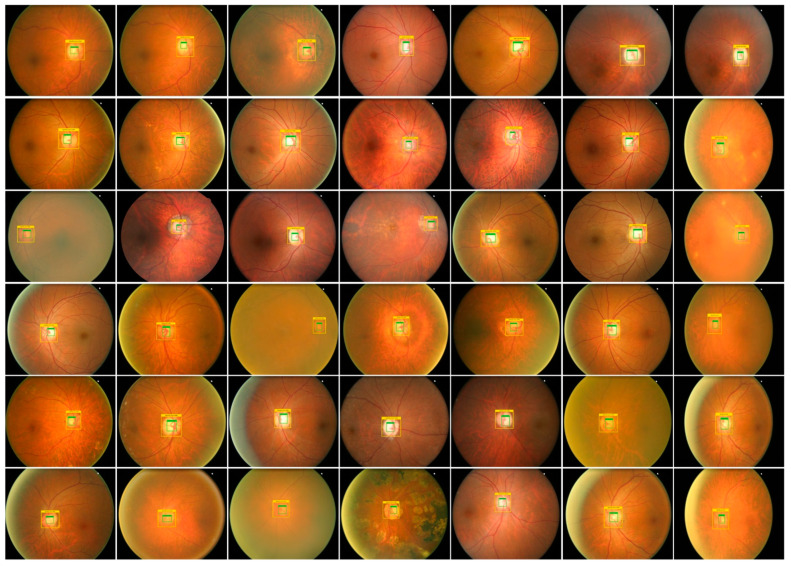
Localization results of EfficientDet-D0 for glaucoma localization.

**Figure 7 sensors-22-00434-f007:**
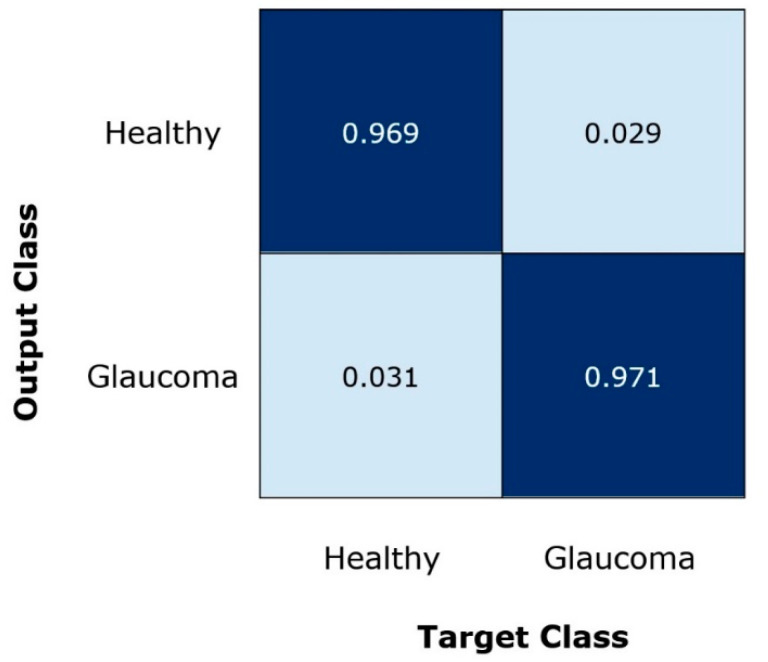
Confusion Matrix of the introduced framework.

**Figure 8 sensors-22-00434-f008:**
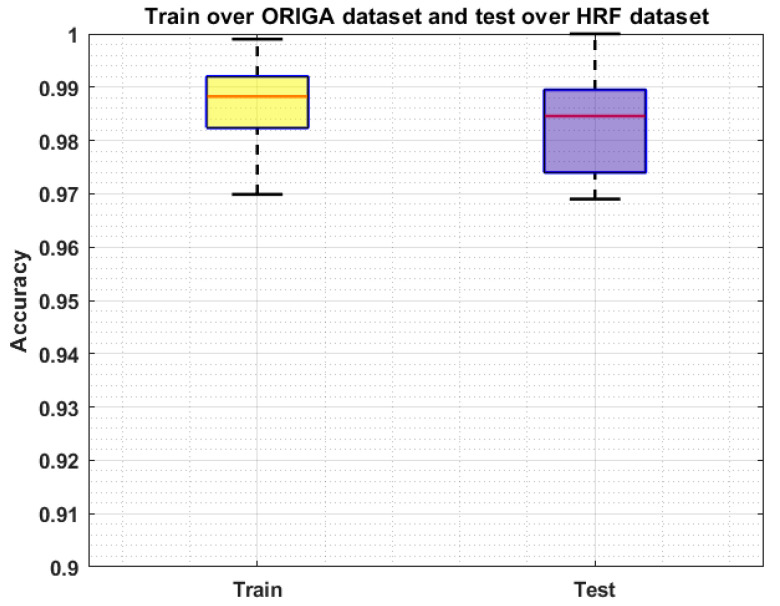
Cross-Validation Results where the model is trained on the ORIGA dataset and test on the HRF dataset.

**Figure 9 sensors-22-00434-f009:**
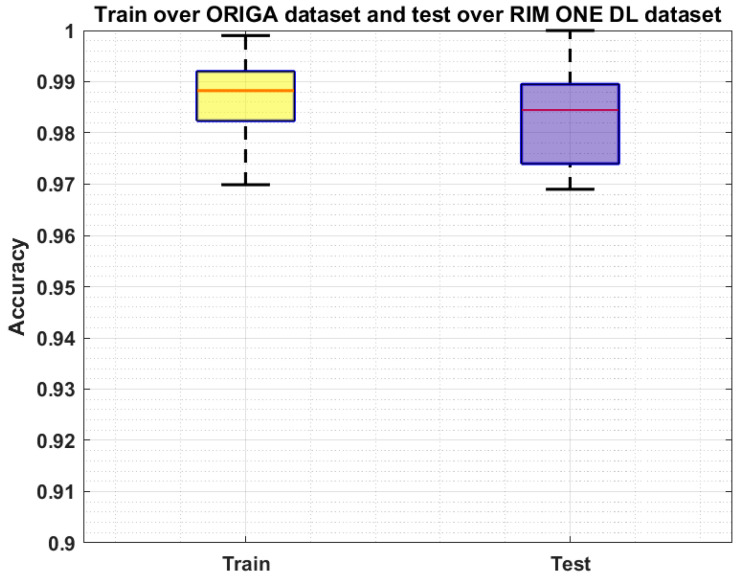
Cross-Validation Results where the model is trained on the ORIGA dataset and test on the RIM ONE DL dataset.

**Figure 10 sensors-22-00434-f010:**
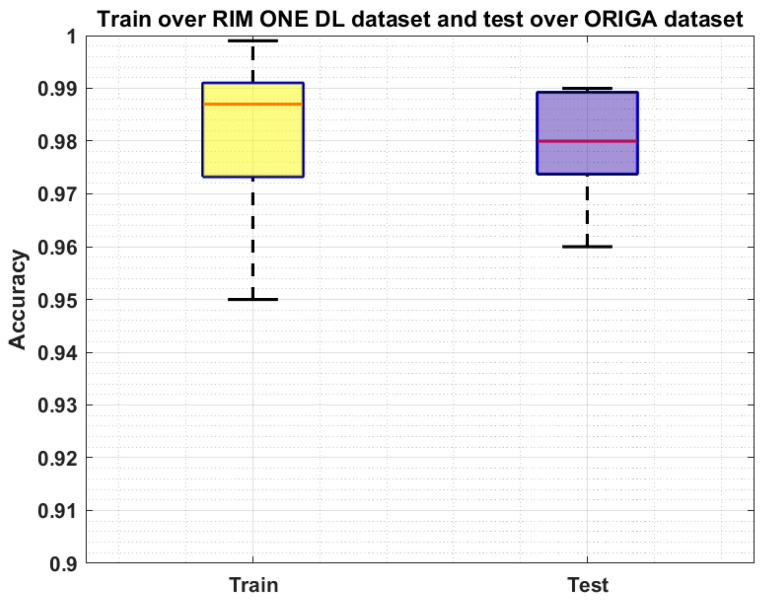
Cross-Validation Results where the model is trained on the RIM ONE DL dataset and test on the ORIGA dataset.

**Table 1 sensors-22-00434-t001:** Comparative analysis of existing approaches.

Reference	Technique	Accuracy	Limitation
ML-based			
[19]	CED, FEM along with the SVM classifier.	93.22%	The model is tested on a small dataset.
[20]	Glowworm Swarm Optimization algorithm	94.86%	The work is unable to compute the cup-to-disc ratio.
[21]	SS-QB-VMD along with the LS-SVM classifier.	92.67%	The classification accuracy requires further improvements.
[22]	Pixel-based threshold along with the watershed transformation	96.1%	The approach is not robust to scale and rotation alterations in the input image.
[23]	The disk selective COSFIRE filters along with the GMLVQ classifier.	97.78%	The work is not robust to noisy samples.
DL-based			
[24]	MobileNetV2 with CNN classifier.	88%	The work requires extensive data for model training.
[25]	ECNet along with the KNN, SVM, BPNN, and ELM classifiers.	96.37%	The technique is economically expensive.
[27]	CNN	98%	The approach needs evaluation on a standard dataset.
[28]	ResNet-50	NA	The work is not robust to noise and blurring in the suspected images.
[29]	DenseNet-201	97%	This approach requires further performance improvements.
[30]	AlexNet, ResNet-50, and ResNet-152	88%	The work requires extensive processing power.
[31]	Mask-RCNN	96.5%	The work needs further performance improvements.
[32]	FRCNN along with the FKM	95%	The work is computationally inefficient.
[33]	UNET	96.44%	Detection accuracy is dependent on the quality of fundus samples.
[34]	VGG-16	83.03%	The model needs extensive training data.
[35]	Faster-RCNN	96.14%	The work is not robust to color variations of the input images.
[36]	WSMTL	NA	The classification performance requires improvements.
[37]	ResNet	88%	The method is not robust to blurry images.

**Table 2 sensors-22-00434-t002:** Training parameters of the proposed solution.

Model Parameters	Value
No. of epochs	60
Learning rate	0.01
Selected batch size	90
Confidence score value	0.5
Unmatched Score value	0.5

**Table 3 sensors-22-00434-t003:** Comparative analysis with other object detection frameworks.

Model	*mAP*	Test Time (s/img)
RCNN	0.913	0.30
Faster-RCNN	0.940	0.25
Mask-RCNN	0.942	0.24
DenseNet77-based Mask-RCNN	0.965	0.23
Proposed	0.971	0.20

**Table 4 sensors-22-00434-t004:** Performance comparison with latest approaches.

Approach	AUC	Recall	Time (s)
Liao et al. [37]	0.880	-	-
Fu et al. [43]	0.910	0.920	-
Bajwa et al. [35]	0.868	0.710	-
Nazir et al. [32]	0.941	0.945	0.90
Nazir et al. [31]	0.970	0.963	0.55
Proposed	0.979	0.970	0.20

**Table 5 sensors-22-00434-t005:** Performance comparison of cross-dataset validation.

Dataset	ORIGA (Test)	HRF (Test)	RIM-ONE DL (Test)
ORIGA (trained)	97.20%	98.21%	97.96%
RIM-ONE DL (trained)	97.83%	98.19%	97.85%

## Data Availability

Not Applicable.

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
