# Peer review of "An Efficient Deep Learning Approach to Automatic Glaucoma Detection Using Optic Disc and Optic Cup Localization"

_sensors, 2022, doi:10.3390/s22020434_

Round 1

Reviewer 1 Report

The authors wrote an article regarding Glaucoma. There are multiple misconceptions of Glaucoma that doesn't make clear the paper. The English level must be improved. The world "Efficient" should be used in a context and with clear and sharp statistically relationship. The sample size it is low to support The conclusions

Author Response

Response sheet is attached. thanks

Reviewer 2 Report

In order to increase the soundness of the paper, the following aspects must be clarified:

  • do not use abbreviations in the title
  • presented methods from related work section must contain also results (accuracy); also the comparison from table 1 must be done based on accuracy results
  • how was chosen the dataset used for evaluation? the proposed method must be evaluated on different other datasets
  • why training and testing time are not compared with methods from table 4 (where accuracy is compared with other methods)?
  • image examples must be added for both the proposed method and the other methods used for comparison in order to provide the better detection of glaucoma.
  • how lesion color, positions and distortions influence the results - add some examples? What distortions are considered? 

Author Response

Response sheet attached. thanks

Reviewer 3 Report

In the manuscript titled "An Efficient Deep learning approach to automatic Glaucoma detection using OD and OC localization", the authors reported that a DL-based approach namely EfficientDet-D0 with EfficientNet-B0 as a glaucoma screening method. EfficientDet-D0 is superior to the latest approaches and more proficient in glaucoma classification. It is interesting to clinical glaucoma diagnosis. The following suggestions should be seriously considered:  

The authors should show patients’ OCT data, because this can show the thickness of the retinal nerve fiber layer (RNFL) and the appearance of optic dics notch, which is an important sign of sensitivity and specificity for glaucoma. This data can enhance and improve AI technology for glaucoma diagnosis. In particular, it could help doctors make difference diagnosis between normal people and early glaucoma patients

Author Response

Response sheet attached. thanks

Reviewer 4 Report

This paper presents an interesting and useful work on developing an automatic glaucoma detection method, based on deep learning approach.  The proposed method combines two blocks by other researchers, which are EfficientNet B0 and BiFPN.  The background of the research has been described well in Section 1.  Section 2 has reviewed mostly recent methods, which is good.  Table 1 provides comparisons between the reviewed methods.  The method has been described well in Section 3.  The authors also have provide good discussions in Section 4.  The number of images used for training and testing are adequate, and obtained from publicly available online dataset.  Comparisons with state-of-the-art methods also have been carried out.  However, there are some minor issues with this paper:

1) First sentence in the abstract.  "Glaucoma is a deadly disease ...", which is not accurate.  Please rewrite.

2) All abbreviations/acronyms should be defined at their first use in the abstract and text.  For example, in line 26, please define what is DL.  Line 29, BiFPN.  Line 33, ORIGA.  Line 34, HRF.  Line 44, OD (although already been defined in the abstract, better to redefine in text).  There are many more.

3) Line 48.  "... where it can be visualized that the blockage in IOF damages the optic nerve."  In my opinion, it would be better if some labels are provided in Figure 1, to show where are the blockage, or damages.

4) Line 50, "... in comparison to a normal human eye."  In my opinion, Figure 1 should also provide a figure showing the eye image from a normal human.

5) Line 58.  " [6-9], However " should be "[6-9].  However".

6) Please check the spelling for EfficientNet.  There are many variations in this paper, such as "EffiicientNet", "EffieicntDet", etc.

7) Line 333.  Please define what is $\phi$ in equation (1).

8) Line 335.  Please write W_bf and D_bf properly (using subscript and in Italic).

9) Figure 6 (a) and (b).  It is not clear on what are the numerator for Precision and Recall.

10) Figure 6, what does the red box means?  What does the black box means?

11) Lne 506.  "H_S.Y." should be "H-S.Y."

Author Response

Response sheet attached. thanks

Round 2

Reviewer 1 Report

The authors extensively improved the paper.

Author Response

Response sheet has been added. thanks

Reviewer 2 Report

Results presented in section 4.6 must be clearly described - a clearly comparison for the proposed solution tested on both datasets must be done.

Author Response

Response sheet has been added. thanks
